# First report of V1016I, F1534C and V410L *kdr* mutations associated with pyrethroid resistance in *Aedes aegypti* populations from Niamey, Niger

**Abdoul-Aziz Maiga[1]\*, Aboubacar Sombié[1], Nicolas Zanré[1], Félix Yaméogo[1], Souleymane Iro[2], Jean Testa[3], Antoine Sanon[1], Ousmane Koita[4], Hirotaka Kanuka[5,6], Philip J. McCall[7], David Weetman[7], Athanase Badolo[1]\***

**1** Laboratoire d'Entomologie Fondamentale et Appliquée, Université Joseph Ki-Zerbo, Ouagadougou, Burkina Faso, **2** Unité de Parasitologie et d'Entomologie Médicale, Centre de Recherche Médicale et Sanitaire, Niamey, Niger, **3** Faculté de Médecine, Université Côte d'Azur, Côte d'Azur, France, **4** Laboratoire de Biologie Moléculaire Appliquée, Université des Sciences, des Techniques et Technologies de Bamako, Bamako, Mali, **5** Center for Medical Entomology, The Jikei University School of Medicine, Tokyo, Japan, **6** Department of Tropical Medicine, The Jikei University School of Medicine, Tokyo, Japan, **7** Department of Vector Biology, Liverpool School of Tropical Medicine, Liverpool, United Kingdom

\* a.badolo@gmail.com (AB); maiga.azizmamadou@gmail.com (AAM)

**Data Availability Statement:** All relevant data are within the manuscript and its Supporting Information files.

## Abstract

### Background

*Ae. aegypti* is the vector of important μ arboviruses, including dengue, Zika, chikungunya and yellow fever. Despite not being specifically targeted by insecticide-based control programs in West Africa, resistance to insecticides in *Ae. aegypti* has been reported in countries within this region. In this study, we investigated the status and mechanisms of *Ae. aegypti* resistance in Niamey, the capital of Niger. This research aims to provide baseline data necessary for arbovirus outbreak prevention and preparedness in the country.

### Methods

Ovitraps were used to collect *Ae. aegypti* eggs, which were subsequently hatched in the insectary for bioassay tests. The hatched larvae were then reared to 3–5-day-old adults for WHO tube and CDC bottle bioassays, including synergist tests. The kdr mutations F1534C, V1016I, and V410L were genotyped using allele-specific PCR and TaqMan qPCR methods.

### Results

*Ae. aegypti* from Niamey exhibited moderate resistance to pyrethroids but susceptibility to organophosphates and carbamates. The *kdr* mutations, F1534C, V1016I and V410L were detected with the resistant tri-locus haplotype 1534C+1016L+410L associated with both permethrin and deltamethrin resistance. Whereas the homozygote tri-locus resistant genotype 1534CC+1016LL+410LL was linked only to permethrin resistance. The involvement of oxidase and esterase enzymes in resistance mechanisms was suggested by partial restoration of mosquitoes' susceptibility to pyrethroids in synergist bioassays.

**Funding:** This work was partially supported in the lab by a WHO/ TDR grant (WHO/TDR/ RCS-KM 2015 ID235974), and the International Collaborative Research Program for Tackling the NTDs Challenges in African countries from Japan Agency for Medical Research and Development, AMED (JP17jm0510002h0003). PJM's research on peri-domestic behavior of Aedes aegypti receives support from MRC-UK (MR/T001267/1). The funders had no role in study design, data collection and analysis, decision to publish, or preparation of the manuscript.

**Competing interests:** The authors have declared that no competing interests exist.

## Conclusion

This study is the first report of *Ae. aegypti* resistance to pyrethroid insecticides in Niamey. The resistance is underpinned by target site mutations and potentially involves metabolic enzymes. The observed resistance to pyrethroids coupled with susceptibility to other insecticides, provides data to support evidence-based decision-making for *Ae. aegypti* control in Niger.

## Introduction

*Aedes aegypti* is a mosquito that spreads several viral diseases including dengue, zika, chikungunya and yellow fever. Among these, dengue and yellow fever are the most important arboviruses affecting West Africa [1]. In recent years, dengue outbreaks and sporadic cases have been reported in the region, notably in Burkina Faso [2], Côte d'Ivoire [3], Senegal [4] and Ghana [5]. Yellow fever epidemics or cases have also occurred in Ghana [6], Nigeria [7] and Côte d'Ivoire [8].

With vaccines lacking, of limited efficacy or in limited supply, the control of these diseases relies on vector control using insecticidal interventions against immature and adult stages in combination with larval source reduction [9,10]. Understanding the resting behaviour (endophily or exophily), blood-feeding patterns and the preferential breeding habitats of immature stages as well as the insecticide resistance status are keys for effective vector control implementation and outbreaks preparedness [11–13]. West Africa is a region where dengue cases are predicted to increase in the coming years [14]. Despite recent reports on *Ae. aegypti* insecticide resistance and bionomics [15], data are still needed in most countries of West Africa to support vector control [16].

Resistance to insecticides is commonly mediated by target site mutations and metabolic enzymes. Target site "knock down resistance" (*kdr*) mutations result in a structural modification of the gene encoding the Voltage Gated Sodium Channel (*Vgsc*) that is targeted by pyrethroid insecticides and DDT [17]. More than 12 *kdr* mutations have been reported in the *Ae. aegypti* VGSC worldwide [18,19]. In West Africa, two *kdr* mutations, V1016I and F1534C have been reported in Cape Verde [20] whereas in Nigeria, S989P and F1534C kdr mutations are reported [21]. Three *kdr* mutations, V410L, V1016I and F1534C have been reported from Burkina Faso [22,23], Côte d'Ivoire [24] and Ghana [25].

Although dengue is not considered endemic in Niger, a first documented imported case during august of 2022 in Niamey [26] should serve as a warning. This is particularly important because neighbouring countries with similar ecological conditions are experiencing dengue outbreaks and regular dengue cases [1]. Vector control remains limited to nationwide regular distribution of long lasting bednets as part of Malaria control strategy [27]. Niger needs to be prepared and to reinforce capacities for early case diagnosis and vector control. This study aimed to document insecticide susceptibility of *Ae. aegypti* from Niamey against common insecticides and investigate the mechanisms which may be involved in resistance, to begin to fill the gap of data on the insecticide resistance in Niger.

## Materials and methods

### Sampling sites

*Ae. aegypti* eggs were collected using ovitraps from 5 sites located in Niamey, the capital city of Niger, from August to October of 2019, during the rainy season. Niamey is located in the

south-western part of the country, has an area of 255 km$^2$ with 1,565,056 inhabitants [28]. The climate is of Sahelian type with a rainy season lasting from June to October and an annual rainfall average of 540 mm. Niamey city is composed of five municipalities or communes and the collections were made at Gamkalé, Kombo, Talladié, INJS and Banifandou 2, situated within three of the municipalities. Gamkalé (13˚29'28.5"N, 002˚07'15.4"E) and Kombo (13˚ 30'45.4"N, 002˚05'51.2"E) are situated in Commune IV of Niamey on the left bank of the Niger River. Talladjé (13˚29'46.8"N, 002˚09'34.0"E) is also situated in Commune IV but not near the river. INJS (13˚30'27.8"N, 002˚05'50.8"E) is located in Commune V on the right bank of the river and Banifandou 2 (13˚32'20.8"N, 002˚08'17.3"E) is situated in Commune II. Gamkalé, Kombo and INJS are characterized by vegetable cropping alongside the river with use of pesticides. By contrast, Talladjé and Banifandou 2 are characterized by high human density and are more urbanized with pesticide use primarily via household insecticides for personal protection against mosquito bites.

**WHO bioassays and CDC bottles tests with synergists.** In total, 106 ovitraps were placed in the gardens or household yards of the 5 sites, after oral informed consent was obtained from the owners of the gardens at Gamkalé & INJS, and from the owners of houses in the remaining three sites.

The ovitraps were made of plastic containers in which filter paper was placed for egg-laying and water from two-day old mango leaf infusion was added. Ovitraps were collected three days later, and filter papers with eggs were dried at room temperature for twenty-four hours, placed in a sealed plastic bag and transported to the insectary of Université Joseph KI-ZERBO, Burkina Faso, for hatching and bioassays. Eggs were pooled across the 5 sites of collection and were hatched in a container with distilled water under controlled insectary conditions of 27.7 ± 1.4˚ C temperature, 79.1 ± 5.5% of relative humidity and a 12: 12 (light: dark) photoperiod. Hatched larvae were reared to adults using Tetramin® and emerged adults were provided with 10% sugar solution until 3 to 5 days old mosquitoes were obtained for bioassay tests.

Due to the absence of WHO-recommended doses for *Ae. aegypti* mosquitoes at the time of bioassays, *Anopheles* diagnostic doses were used [29]. Though these doses are slightly higher than *Aedes* doses and so provide a conservative assessment of resistance, they have been commonly used [13,18,25,30]. Insecticide impregnated papers including permethrin 0.75%, deltamethrin 0.05%, Malathion 5%, Pirimiphos-Methyl 0.25% and Bendiocarb 0.1% ordered from the WHO reference Center in Malaysia, were tested against *Ae. aegypti* populations. For each insecticide test, four tubes of 20–25 female mosquitoes of 3 to 5 days old and two control tubes were used. After one hour exposure to insecticide, mosquitoes were removed from exposure tubes to observation tubes and kept for 24 hours. The number of mosquitoes dead after 24 hours was counted and the mortality calculated and corrected with Abbott's formula [31] when mortality in the control was between 5 to 20%. Survivors and dead mosquitoes from bioassays were stored in 1.5 ml tubes over silica gel and kept at– 20˚C for molecular analysis. Bioassays were performed under the same temperature and humidity conditions as mosquito rearing (27.7 ± 1.4˚ C temperature, 79.1 ± 5.5% of relative humidity).

Bottle bioassays with and without synergists were also performed to assess the effect of synergist pre-exposure on pyrethroid insecticide efficacy and to assess the possible involvement of metabolic enzymes in pyrethroid resistance. Susceptibility tests were performed following the CDC bottle bioassay guidelines [32], but with modifications according to the WHO guidelines. [33] Insecticide stock solutions were prepared using acetone as solvent, at concentrations of 15 µg/ml for permethrin and 10 µg/ml for deltamethrin. Synergist stock solutions were prepared at concentrations of 400 µg/ml for PBO (Piperonyl butoxide) and 125 µg/ml for DEF (S. S.S-tributylphosphorotrithioate). From each insecticide and synergist solution, 1 ml per bottle

was used to coat 4 bottles of 250 ml for the insecticide and 2 control bottles coated with acetone only. Coated bottles were dried at room temperature for 24 hours and kept in the fridge until use. For each bioassay using synergist, 100–125 non-blood-fed female mosquitoes of 3–5 days old were first exposed to synergists (synergist exposure bottles) at a rate of 25 mosquitoes per bottle for 1 hour, then transferred to holding cups before transfer to 4 replicate pyrethroid-coated bottles and one acetone-coated bottle as a negative control for 1 hour exposure [32,33]. In parallel, direct insecticide exposure bioassay tests were done with 100–125 mosquitoes first exposed to acetone (as synergist control bottles) for 1 hour, then transferred to holding cups before transfer to 4 replicate pyrethroid insecticides coated bottles for 1 hour and one acetone coated bottle as negative control. After exposure, the mortality was recorded 24h after, and Abott's correction was made if the mortality rate in the control was between 3 and 10%. Dead and alive mosquitoes were kept in 1.5 ml tube over silica gel and stored at– 20˚C. Bottle bioassays tests were done under the same controlled insectary conditions as the tube bioassays (27.7 ± 1.4˚ C temperature, 79.1 ± 5.5% of relative humidity).

## DNA extraction

Each mosquito was homogenised in 100 µl of buffer A (containing 0.1M tris pH 9.0, 0.1 M EDTA, 1% Sodium DodécylSulfate (SDS) and 0.5% Diethyl Pyrocarbonate (DEPC) with a sterilized pellet mixer. The homogenate was incubated at 70˚ C for 30 min in a block incubator (Labnet Dry Bath, Dual block, Ref D1302), then 22.4 µl of 5M of potassium acetate (KoAc) was added. The mixture was vortexed and cooled on ice for 30 min. The mixture was centrifuged for 15.000 rpm at 4˚ C for 15 min, after which 90 µl of supernatant was collected and 45 µl of isopropanol added. After vortexing, the mixture was centrifuged at 15,000 rpm at 4˚ C for 20 min. After centrifugation, the supernatant was discarded, and the pellet was rinsed with 200 µl of 70% ethanol. This mixture was centrifuged for 15.000 rpm at 4˚ C for 5 min, the supernatant discarded. The pellet was dried before being dissolved in 50 µl of Tris-EDTA pH 8.0 buffer.

## Allele-specific—PCR for the detection of F1534C, V1016I and V410L mutations

A total of 202 mosquitoes including alive and dead from permethrin 0.75% and deltamethrin 0.05% were chosen for genotyping the F1534C, V1016I and V410L *kdr* mutations by using AS-PCR methods. Allele Specific PCR procedures for the detection of V1016I and F1534C *kdr* mutations used the conditions detailed in Sombié et al. [34]. Detection of the V410L *kdr* mutation was performed following the protocol of Granada et al. [35] with modifications in the PCR conditions described in Sombié et al [22].

Two PCRs were used to genotyping the V and L alleles of the V410L *kdr* mutation using the primers listed in S1 Table. Each PCR was performed with a reactional volume of 12.5 including 1µl of Target DNA, 2.5 µl of primer's mix (0.3 µM), 2.75 µl of sterile water and 6.25 µl of taq polymerase mix (AmpliTaq Gold®, Master mix, thermo Fischer Scientific). The PCR conditions were: initial denaturation for 7 minutes at 95˚C, 35 cycles of extension at 95˚C for 30s, 60˚C for 30s and 72˚C for 1-minute, terminal elongation at 72˚C for 7 minutes.

## TaqMan detection of F1534C, V1016I and V410L *kdr* mutations

A total of 50 specimens of F0 mosquitoes (i.e., raised from field collected eggs), were screened using TaqMan qPCR to estimate the *kdr* mutations frequencies in the population. Reactions were performed in 96 well plates by adding 5 µl of TaqMan gene expression SensiMix (Applied Biosystem, Foster City, USA), 0.125 µl of primer/probe, 3.875 µl of molecular grade sterile water and 1 µl of the DNA extract [13]. Reactions were run in a separate reaction for each *kdr*

locus on an Agilent MX3000P qPCR thermal cycler using cycling conditions of an initial denaturation of 10 min at 95° C, followed by 40 cycles of 92° C for 15 min and 60° C for 1 min.

## Statistical analysis

Mosquito susceptibility to insecticides in WHO tubes bioassay was interpreted according to WHO guidelines [29]. Mortality rates between 98–100% indicated susceptibility, mortality rates from 90 to 97% indicated possible resistance and mortality rates less than 90% indicated confirmed resistance. The mortalities of bottles bioassay with synergist were also interpreted according to WHO guidelines [33]: ≥98% mortality indicates full restoration of susceptibility; <98% but ≥90% increase in mortality indicates partial effect of synergist and enzyme involvement; < 90% mortality indicates no effect of synergist and no involvement of enzymes targeted. We used Generalized Linear Models (GLM) with binomial link function [36] using R package "stats version 4.0.5" to assess the effect of insecticide and their synergists on the mortality of *Ae. aegypti*. *Kdr* allele and genotype frequencies were calculated and Fisher's exact two-sided test in R software version 4.0.5 was used to assess the association between *kdr* individual, tri-locus genotypes and mortality.

## Results

### *Ae. aegypti* susceptibility to insecticides in Niamey

WHO bioassays showed that *Ae. aegypti* mosquitoes from Niamey were susceptible to malathion and bendiocarb with 100% mortality rates, and with mortality rate of 99% to pirimiphos-methyl [CL:97.04–100]. Conversely, the mosquitoes were resistant to permethrin and deltamethrin with mortality rates of 82.52% [CL: 76.06–88.98] and 83.26% [CL: 80.64–85.8], respectively (Fig 1 and S2 Table).

### Alleles and genotypes of *kdr* mutations and their association with pyrethroid resistance

*Kdr* mutant allele frequencies were 0.30 for 1534C and 0.04 for both 1016I and 410Lin the cohort genotyped using TaqMan. Those in the cohort exposed to pyrethroids were

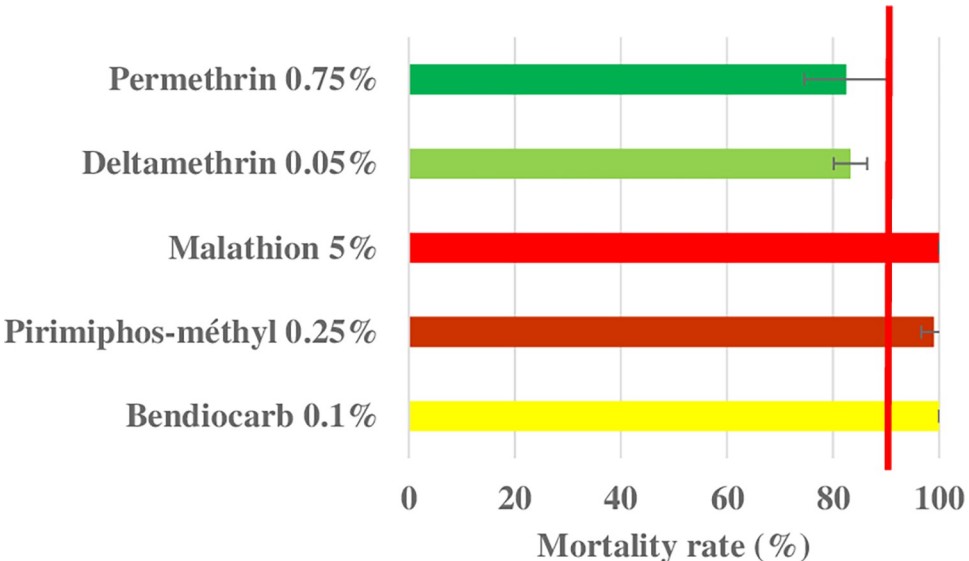

**Fig 1. WHO tubes bioassay mortality of *Ae. aegypti* exposure to insecticides.** The red line indicates the WHO resistance thresholds. Error bars represent 95% confidence interval of the mean.

**Table 1. Genotypes and, *kdr* allele frequencies of F1534C, V1016I and V410L mutations with 95% confidence intervals detected by TaqMan in F0 *Ae. aegypti* and by Allele Specific PCR in F1&F2 mosquitoes used in bioassays.**

| Method | Number of mosquitoes | F1534C genotype | | | Freq. of C allele | V1016I genotype | | | Freq. of I allele | V410L genotype | | | Freq. of L allele |
|---|---|---|---|---|---|---|---|---|---|---|---|---|---|
| | | CC | FC | FF | | II | VI | VV | | LL | VL | VV | |
| TaqMan | 50 | 8 | 14 | 28 | 0.3 | 00 | 4 | 46 | 0.04 | 00 | 4 | 46 | 0.04 |
| C.I. | | | | | 0.21–0.40 | | | | 0.01–0.10 | | | | 0.01–0.10 |
| AS-PCR | 202 | 48 | 111 | 43 | 0.51 | 4 | 53 | 145 | 0.15 | 5 | 35 | 162 | 0.11 |
| C.I. | | | | | 0.46–0.56 | | | | 0.12–0.19 | | | | 0.08–0.15 |

significantly higher, but the same alleles were detected in each assay, and the frequency differences likely reflect inter-cohort differences (Tables 1 and S6). Ten genotypes were found out of 27 possible genotype combinations across the three *kdr* mutations F1534C, V1016I and V410L based on AS-PCR assay (Fig 2). The most common genotype was the single 1534 mutant FC/VV/VV at 39.6%, the triple-homozygote mutant for the three mutations CC/II/LL was found at around 2% while the triple homozygote wild-type FF/VV/VV was recorded at 18.8%. From these genotypes, six haplotypes were recorded with relative proportions of: FVV (47.5%), CVV (37.1%), CIL (10.9%), CIV (3%), FIV (1.2%) and CVL (0.2%) (Fig 3).

The tri-locus genotypes CC/VI/VL, CC/II/LL and FC/VI/VV were significantly associated with permethrin survival compared to the reference wild type genotype FF/VV/VV for which there were no survivors (Tables 2 and S3). The tri-locus haplotypes CIL, CVV and CIV were significantly associated with permethrin resistance with respectively $p < <0.001$, $p = 0.014$ and $p = 0.016$. (S4 and S3 Tables).

The tri-locus genotypes CC/VV/VV, CC/VI/VL and CC/VI/LL showed significant association with deltamethrin resistance compared to the reference wild type genotype FF/VV/VV for which again there were no survivors (Tables 3 and S3). Unlike the permethrin dataset, the homozygote resistant genotype CC/II/LL was not present in any mosquitoes tested with deltamethrin. The tri-locus haplotypes CVV and CIL were significantly associated with deltamethrin resistance with respectively $p << 0.001$ and $p < 0.001$ (S3 and S5 Tables).

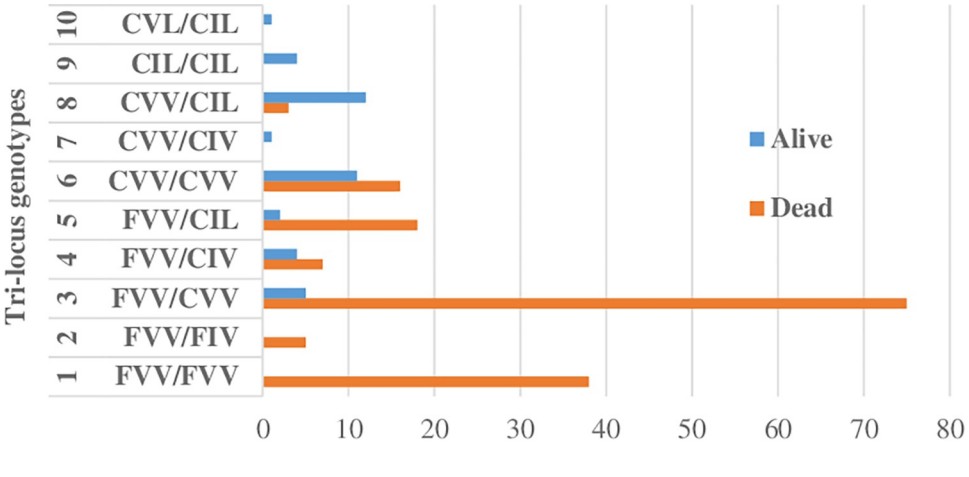

**Fig 2. Number of tri-locus genotype combination of F1534C, V1016I and V410L *kdr* mutations in dead and alive pyrethroid-exposed *Ae. aegypti*.**

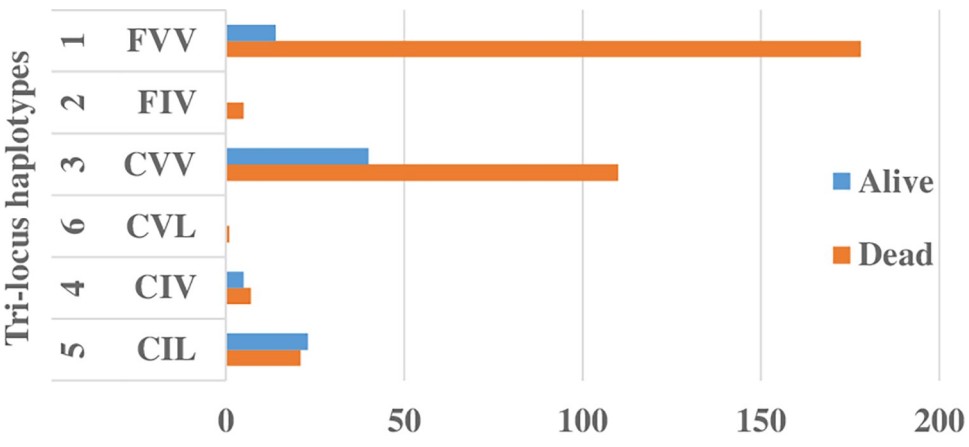

**Fig 3. Number of tri-locus haplotype of F1534C, V1016I and V410L *kdr* mutations in dead and alive pyrethroid-exposed *Ae. aegypti*.**

## Effect of synergists on pyrethroid efficacy

Pre-exposure to synergists increases mortality to permethrin and deltamethrin in *Ae. aegypti* from Niamey. Exposure to diagnostic concentrations of permethrin and deltamethrin resulted in 67.41% and 75.91% mortality, respectively (Fig 4 and S6 Table). Both PBO and DEF synergists significantly increased susceptibility to permethrin with respective mortality rates of 99.03% (p < 0.001) and 95.21% (p < 0.001) (Table 4 and Fig 4). Whilst PBO completely restored susceptibility to permethrin, DEF enhanced mortality rate but was still below the susceptibility threshold, suggesting a partial restoration of susceptibility. In contrast, only PBO significantly increased mosquito susceptibility with deltamethrin (mortality rate from 75.91% to 97.5%, p = 0.001) with a partial restoration of susceptibility, whereas DEF did not significantly increase mosquito susceptibility to deltamethrin with a mortality rate of 82.22% (p = 0.30) (Table 4 and Fig 4).

**Table 2. Genotypes and their association with resistance to permethrin of *Aedes aegypti*.**

| Number | Genotypes | Phenotypes | | Fisher exact P-value |
|---|---|---|---|---|
| | | Dead (Susceptible) | Alive (Resistant) | |
| 1 | FF/VV/VV | 16 | 0 | Reference |
| 2 | FF/VI/VV | 4 | 0 | 1 |
| 3 | FC/VV/VV | 35 | 2 | 0.570 |
| **4** | **FC/VI/VV** | 6 | 3 | **0.037** |
| 5 | FC/VI/VL | 10 | 0 | 1 |
| 6 | CC/VV/VV | 9 | 3 | 0.067 |
| 7 | CC/VI/VV | 0 | 1 | 0.059 |
| **8** | **CC/VI/VL** | 1 | 7 | **<0.001** |
| **9** | **CC/II/LL** | 0 | 4 | **<0.001** |
| 10 | CC/VI/LL | 0 | 0 | - |

Significantly associated genotypes are highlighted in bold.

**Table 3. Genotypes and their association with resistance to deltamethrin of *Ae. aegypti*.**

| Number | Genotypes | Phenotypes | | Fisher exact P-value |
|---|---|---|---|---|
| | | Dead (Susceptible) | Alive (Resistant) | |
| 1 | FF/VV/VV | 22 | 0 | Reference |
| 2 | FF/VI/VV | 1 | 0 | 1 |
| 3 | FC/VV/VV | 40 | 3 | 0.318 |
| 4 | FC/VI/VV | 1 | 1 | 0.083 |
| 5 | FC/VI/VL | 8 | 2 | 0.091 |
| 6 | **CC/VV/VV** | 7 | 8 | **<0.001** |
| 7 | CC/VI/VV | 0 | 0 | - |
| 8 | **CC/VI/VL** | 2 | 5 | **<0.001** |
| 9 | CC/II/LL | 0 | 0 | - |
| 10 | **CC/VI/LL** | 0 | 1 | **0.043** |

Significantly associated genotypes are highlighted in bold.

## Discussion

This study reports for the first time the resistance to pyrethroid insecticides in *Ae. aegypti* populations from Niamey, Niger. Three *kdr* mutations (F1534C, V1016I, and V410L) have been recorded, and oxidase and esterase detoxification enzymes (inferred from synergist assays) appear to be associated with the pyrethroid resistance phenotypes. The population was found to be susceptible to carbamate and organophosphate insecticides and these might represent useful options for insecticidal spray interventions in Niamey. Though the study is limited to Niamey, the presence of multiple insecticide resistance mechanisms in *Ae. aegypti* mosquito populations is a concern for vector control programs in this urban setting where a dengue outbreak is perhaps most likely to occur. A resistance surveillance program is needed here for *Ae. aegypti* control, to help toward preparedness for dengue prevention and outbreaks.

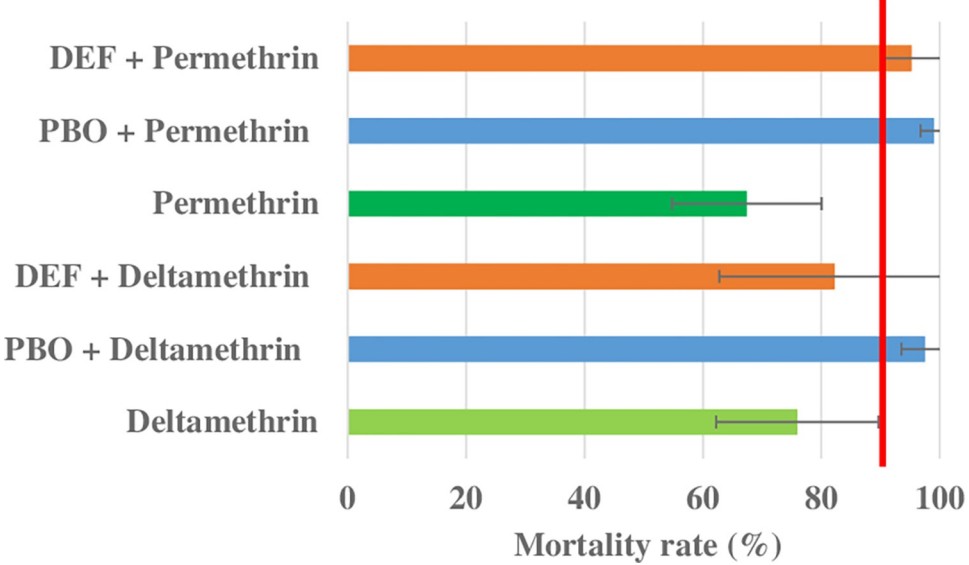

**Fig 4. Bottle bioassay mortality of *Ae. aegypti* exposed to deltamethrin (10 μg/ml) and permethrin (15 μg/ml) insecticides with PBO and DEF synergists.** The red line indicates the resistance thresholds. Error bars represent 95% confidence interval of the mean.

**Table 4. Generalised linear model of the mortality of *Ae. aegypti* exposure to insecticides and synergists.**

| Predictors | Estimate | 95% CL | z-value | Pr(>\|z\|) |
|---|---|---|---|---|
| **Intercept** | **0.736** | **[0.328–1.145]** | **3.531** | **<0.001** |
| **Insecticide [Permethrin]** | | | | |
| **Permethrin + DEF** | **2.327** | **[1.342–3.313]** | **4.628** | **<0.001** |
| Permethrin + PBO | **3.859** | **[1.847–5.871]** | **3.759** | **<0.001** |
| | | | | |
| **Intercept** | **1.173** | **[0.733–1.613]** | **5.225** | **<0.001** |
| **Insecticide [Deltamethrin]** | | | | |
| **Deltamethrin + PBO** | **2.541** | **[1.071–4.011]** | **3.388** | **0.001** |
| **Deltamethrin + DEF** | 0.356 | [-0.317–1.029] | 1.036 | 0.300 |

Reference factor levels of predictors are shown in brackets, with beta and effect size estimates, confidence intervals, z-value and associated probabilities for predictors included in the model. Significant predictor terms are shown in bold.

Though *Ae. aegypti* has not been a specific target of vector control, insecticide use in urban agriculture, and in domestic environments, may have played a role in resistance development. The localities of Gamkallé, Kombo and INJS are situated near the Niger River where vegetables are cultivated with pesticides and fertilizers used for crops protection. That may play some role in the development of insecticide resistance, but given the adaptation of *Ae. aegypti to* artificial breeding containers this is probably of limited importance [37]. There is no history of indoor residual spraying in Niger, and only insecticide treated bednets have been scaled up as malaria control strategy with bednet distribution every three years and the first largest distribution in 2006 [38]. Despite the exophilic behaviour of *Ae. aegypti* in West Africa [12,39,40], bednet exposure may also have played a role in the development of *Ae. aegypti* insecticide resistance as has been the case for malaria vectors [27]. Domestic use of insecticide such as aerosols or coils to deter mosquito bites may play a more important role in the selection of insecticide resistance in mosquito populations [37]. However, in Niger and in West Africa generally sources of selection leading to resistant *Ae. aegypti* require further investigation.

The allele frequency of 1534C found in Niamey remains lower than elsewhere in West African studies from Ouagadougou [22], Abidjan [24] and Accra [25] where it is at or approaching fixation, similar to Latin American countries such as Colombia [35] and Mexico [41]. The 1016I allele frequency was also much lower in this study than in Angola [20], Abidjan [24], Accra [25], Ouagadougou [22] and Mexico [41]. Similarly, the frequency of 410L was lower than found in recent studies from Ouagadougou [22] and Abidjan [24], though frequencies in Accra [25] and Colombia [35] remain relatively low. In contrast, highest frequencies were reported from Angola [20] and Mexico [42]. Alone, or in combination with F1534C and V1016I, the V410L mutation can confer resistance to pyrethroids [42]. The V1016I and V410L mutations appear linked in *Aedes* populations from Niger. Similar trend has been found in Latin America and in Burkina Faso [22,42]. However, in Ghana [25] the dissociation appears much greater suggesting a different stage of co-evolution for the two mutants [42].

The triple homozygote resistant CIL/CIL was found to be associated with both permethrin and deltamethrin resistance. Similar findings have been reported from Mexico [42], whereas in Ghana this tri-locus genotype has been only associated with permethrin resistance [25]. In this study, the double-locus heterozygote genotype CVV/CIL was found to be associated with both permethrin and deltamethrin resistance, whereas it was associated only with deltamethrin resistance in Mexico [42]. We also found the CIL haplotype to be associated with both permethrin and deltamethrin resistance. In contrast, in Ouagadougou it was associated only with permethrin resistance [22].

The diversity of tri-locus genotypes (10) and tri-locus haplotypes (6) that we found suggest a late emergence of the insecticide resistance compared to Ouagadougou in Burkina Faso where the 1534C mutation is almost fixed, effectively reducing the number of genotypes (6) and haplotypes (3) and to the three mutations [22].

Pre-exposure to the synergist PBO significantly restored susceptibility to both permethrin and deltamethrin insecticides suggesting probable involvement of P450 family genes in the resistance phenotypes [13,37]. Pre-exposure to the synergist DEF also significantly elevated susceptibility to permethrin, though the rise in mortality with deltamethrin exposure was not significant. This is suggestive of the role of esterase family enzymes in pyrethroid resistance [43]. Whilst both target site and metabolic resistance appear to underly pyrethroid resistance in Niamey, metabolic resistance appears to be the main mechanism of resistance in multiple populations from Senegal in which *kdr* mutants were absent [44]. Further studies are required to investigate the spatial extent of resistance/susceptibility profiles across insecticides in Niger, and to determine whether similar resistance mechanisms apply more widely.

## Conclusion

We reported for the first-time multiple insecticide resistance mechanisms including target site mutations and metabolic enzymes among *Ae. aegypti* population from Niamey, the largest city of Niger. Our study implicates three target site mutations (F1534C + V1016I + V410L) and metabolic enzymes (oxidases and esterases) as mechanisms involved in resistance to pyrethroids, whilst highlighting susceptibility to alternate insecticide classes. This study provides both important and baseline data for *Ae. aegypti* borne disease control and for vector insecticide resistance monitoring respectively in Niamey, Niger.

## Supporting information

**S1 Table. List of primers sequences used for detecting V410L *kdr* mutation.**
(DOCX)

**S2 Table. 1 hour WHO tubes bioassay data.**
(DOCX)

**S3 Table. 1 hour bottles bioassay data.**
(DOCX)

**S4 Table. Haplotypes and their association with resistance to permethrin of *Ae. aegypti*.**
Significantly associated haplotypes are highlighted in bold.
(DOCX)

**S5 Table. Haplotypes and their association with resistance to deltamethrin of *Ae. aegypti*.**
Significantly associated haplotypes are highlighted in bold.
(DOCX)

**S6 Table. Genotypes from AS-PCR and Taqman detection of the kdr mutations.**
(DOCX)

## Acknowledgments

We would like to thank the populations from the 5 collection sites for their cooperation during mosquito sampling.

## Author Contributions

**Conceptualization:** Abdoul-Aziz Maiga, Philip J. McCall, David Weetman, Athanase Badolo.

**Data curation:** Abdoul-Aziz Maiga, Aboubacar Sombié, Jean Testa, David Weetman, Athanase Badolo.

**Formal analysis:** Abdoul-Aziz Maiga, Aboubacar Sombié, Jean Testa, David Weetman, Athanase Badolo.

**Funding acquisition:** Hirotaka Kanuka, Philip J. McCall, David Weetman, Athanase Badolo.

**Investigation:** Abdoul-Aziz Maiga, Nicolas Zanré, Félix Yaméogo, Souleymane Iro, Athanase Badolo.

**Methodology:** Abdoul-Aziz Maiga, Aboubacar Sombié, Philip J. McCall, David Weetman, Athanase Badolo.

**Project administration:** Antoine Sanon, Athanase Badolo.

**Resources:** Antoine Sanon, Hirotaka Kanuka, Athanase Badolo.

**Software:** Aboubacar Sombié, Jean Testa, Athanase Badolo.

**Supervision:** Ousmane Koita, David Weetman, Athanase Badolo.

**Validation:** Aboubacar Sombié, David Weetman, Athanase Badolo.

**Visualization:** Abdoul-Aziz Maiga, Aboubacar Sombié, David Weetman, Athanase Badolo.

**Writing – original draft:** Abdoul-Aziz Maiga, Athanase Badolo.

**Writing – review & editing:** Aboubacar Sombié, Nicolas Zanré, Félix Yaméogo, Souleymane Iro, Jean Testa, Antoine Sanon, Ousmane Koita, Hirotaka Kanuka, Philip J. McCall, David Weetman, Athanase Badolo.

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
