## [Decision Letter · Decision Letter 0]

12 Sep 2023

PONE-D-23-20467First report of V1016I, F1534C and V410L kdr mutations associated with pyrethroid resistance in Aedes aegypti populations from Niamey, NigerPLOS ONE

Dear Dr. Badolo,

Thank you for submitting your manuscript to PLOS ONE. After careful consideration, we feel that it has merit but does not fully meet PLOS ONE’s publication criteria as it currently stands. Therefore, we invite you to submit a revised version of the manuscript that addresses the points raised during the review process.

We look forward to receiving your revised manuscript.

Kind regards,

Nicholas C. Manoukis

Academic Editor

PLOS ONE

Journal Requirements:

a) The name of the colleague or the details of the professional service that edited your manuscript.

b) A copy of your manuscript showing your changes by either highlighting them or using track changes (uploaded as a *supporting information* file).

c) A clean copy of the edited manuscript (uploaded as the new *manuscript* file).

3. Thank you for stating the following in your Competing Interests section:  "NO authors have competing interests"

6. We note that Figure 1 in your submission contain map images which may be copyrighted. All PLOS content is published under the Creative Commons Attribution License (CC BY 4.0), which means that the manuscript, images, and Supporting Information files will be freely available online, and any third party is permitted to access, download, copy, distribute, and use these materials in any way, even commercially, with proper attribution. For these reasons, we cannot publish previously copyrighted maps or satellite images created using proprietary data, such as Google software (Google Maps, Street View, and Earth). For more information, see our copyright guidelines: http://journals.plos.org/plosone/s/licenses-and-copyright.

(1) You may seek permission from the original copyright holder of Figure 1 to publish the content specifically under the CC BY 4.0 license.  

**Additional Editor Comments:**

Manuscript should be carefully reviewed and many parts significantly overhauled. All reviewers note questions on methodology- these must be clarified/answered as they are critical for interpretability/repeatability.

Reviewers' comments:

Reviewer's Responses to Questions

**Comments to the Author**

1. Is the manuscript technically sound, and do the data support the conclusions?

Reviewer #1: Partly

Reviewer #2: Partly

Reviewer #3: Yes

2. Has the statistical analysis been performed appropriately and rigorously? 

Reviewer #1: I Don't Know

Reviewer #2: I Don't Know

Reviewer #3: Yes

3. Have the authors made all data underlying the findings in their manuscript fully available?

Reviewer #1: Yes

Reviewer #2: Yes

Reviewer #3: Yes

4. Is the manuscript presented in an intelligible fashion and written in standard English?

Reviewer #1: Yes

Reviewer #2: No

Reviewer #3: Yes

5. Review Comments to the Author

Reviewer #1: 1. Introduction: 'Data on Aedes aegypti bionomics and insecticide resistance status are still lacking' - There have been various insecticide resistance reports in Aedes aegypti in West Africa. 

2. Methods: Bioassay tests were performed according to World Health Organizations (WHO) protocol using Anopheles diagnostic doses [27] - Justify why Anopheles diagnostic doses were used when Aedes diagnostic doses are available for resistance testing. 

3. Methods: Temperature and humidity during the tests were not stated.

4. Methods: How many samples were used for Allele-specific - PCR for the detection of F1534C, V1016I and V410L mutation

5. Methods: What were the modifications done on the protocol of Granada et al.  2018?

6. Methods, Results, Discussion: 'Elevated activity of oxidase and esteraseenzymes were detected and may also be involved in metabolic resistance to pyrethroids' was mentioned in the abstract but the methodologies and results were not mentioned and discussed. The elevation of activity should not be based on a synergist test. 

7. Results: 'A total of 10 tri-locus combinations genotypes were found out of 27 possible (Figure 4).' - Rewrite to clarify.

8. What was the history of insecticide use in the study areas?

Reviewer #2: 1. Please add numbered lines throughout the manuscript; pointing out the reviewer's comments is difficult.

2. Methodology

CDC bottle bioassays interpretations

It would be necessary to update the interpretation of the resistance status based on the % mortality in the bottle bioassays. Since the authors refer to the WHO 2016 manual for selecting DD, in the same manual, the interpretation of the results in bottle and cylinder of the WHO is established in the same way.

The methodology for detecting kdr mutations is confusing. Please clarify. AS-PCR should be detailed. Please describe the primers used. Are you comparing two methods?

Which mosquitoes were submitted to each method? You said that 50 non-exposed mosquitoes were genotyped by qPCR as described in the methodology but in AS-PCR, what type of mosquitoes were used, and how many?

What do you mean by exposed mosquitoes, alive or dead?

What do you mean by non-exposed mosquitoes, field mosquitoes?

Please explain it the detailed in the methodology.

3. Results

It is confusing the way of expressing the kdr genotypes. Please use two alleles for each locus.

The tri-locus haplotypes CIL, CVV, and CIV were significantly associated with permethrin resistance ( p < 10-6 , p = 0.014 and p = 0.016, respectively) (S3 table).

Those results are in Table 2.

Discussions are limited to investigations done on the continent. There is a vast literature on the association of kdr mutations and pyrethroid resistance worldwide that should be included.

Reviewer #3: 1. The authors state bioassays were conducted as per Ref #29 - Guideline for Evaluating Insecticide Resistance in Vectors Using the CDC Bottle Bioassay. While the stated diagnostic was used for the CDC bottle bioassays, the authors state that mosquitoes were exposed to the insecticides for 1 hour while the diagnostic timepoint for the CDC bottle bioassays at the diagnostic concentration used is stated as 30 minutes in the reference. The authors do not explain why this modification was made in the submitted manuscript.

2. Throughout the manuscript - the authors alternate between "Ae aegypti" and "Aedes aegypti" - please consider improving consistency

3. Knockdown should be defined.

4. Clarification is required on whether knockdown was recorded for a total of 60 minutes throughout the exposure period or if knockdown was recorded after the 60 min exposure.

5- Allele specific PCR - please specify what modifications were made to the published protocol.

6- Statistical analysis - "Mortalities" should be "mortalities"

7. Major consistency and formatting errors in the references section.

6. PLOS authors have the option to publish the peer review history of their article (what does this mean?). If published, this will include your full peer review and any attached files.

Reviewer #1: No

Reviewer #2: No

Reviewer #3: No

---

## [Author Response · Author response to Decision Letter 0]

29 Mar 2024

All responses to reviewers comments are included in the cover and rebuttal letter

---

## [Decision Letter · Decision Letter 1]

8 May 2024

PONE-D-23-20467R1First report of V1016I, F1534C and V410L kdr mutations associated with pyrethroid resistance in Aedes aegypti populations from Niamey, NigerPLOS ONE

Dear Dr. Badolo,

Thank you for submitting your manuscript to PLOS ONE. After careful consideration, we feel that it has merit but does not fully meet PLOS ONE’s publication criteria as it currently stands. Therefore, we invite you to submit a revised version of the manuscript that addresses the points raised during the review process.

We look forward to receiving your revised manuscript.

Kind regards,

Nicholas C. Manoukis

Academic Editor

PLOS ONE

Journal Requirements:

Additional Editor Comments:

Thank you for addressing the comments, I believe the paper is much improved. The last remaining issue is deposit of the study data - which some aggregate information from resistance assays is provided it is not in primary form (by replicate) and I was not able to find allele typing data in the SI (ex. data of alleles for each locus from each of the 50 specimens screened via TaqMan assay- I think only aggregate is provided in Table 1). I believe it would be appropriate to deposit all the raw data from the study in an online database and cite it in the paper.

Table 2 seems to have some typesetting issues in the pdf.

Reviewers' comments:

Reviewer's Responses to Questions

**Comments to the Author**

1. If the authors have adequately addressed your comments raised in a previous round of review and you feel that this manuscript is now acceptable for publication, you may indicate that here to bypass the “Comments to the Author” section, enter your conflict of interest statement in the “Confidential to Editor” section, and submit your "Accept" recommendation.

Reviewer #1: All comments have been addressed

2. Is the manuscript technically sound, and do the data support the conclusions?

Reviewer #1: Yes

3. Has the statistical analysis been performed appropriately and rigorously? 

Reviewer #1: Yes

4. Have the authors made all data underlying the findings in their manuscript fully available?

Reviewer #1: Yes

5. Is the manuscript presented in an intelligible fashion and written in standard English?

Reviewer #1: Yes

6. Review Comments to the Author

Reviewer #1: The revision for PONE-D-23-20467R1, entitled "First report of V1016I, F1534C and V410L kdr mutations associated with pyrethroid resistance in Aedes aegypti populations from Niamey, Niger"is satisfactory. Thank you for addressing my comments

7. PLOS authors have the option to publish the peer review history of their article (what does this mean?). If published, this will include your full peer review and any attached files.

Reviewer #1: No

---

## [Author Response · Author response to Decision Letter 1]

9 May 2024

- We have revised all the reference and we noticed that the reference Moyes et al (2017) has been corrected in 2021, then we have replaced the old reference by the new and correct one.

- We provided in a supplementary table the raw data of the genotypes recorded from the AS-PCR and Taqman methods. We expected the entire data are made now available to the readers.

-We have revised the table 2 and 3 and the typesetting issues in the pdf are solved.

- We appreciate the positive feedback from the reviewer who recognised our effort to address his comments in the previous version of the manuscript.

---

## [Editor Report · Decision Letter 2]

15 May 2024

First report of V1016I, F1534C and V410L kdr mutations associated with pyrethroid resistance in Aedes aegypti populations from Niamey, Niger

PONE-D-23-20467R2

Dear Dr. Badolo,

We’re pleased to inform you that your manuscript has been judged scientifically suitable for publication and will be formally accepted for publication once it meets all outstanding technical requirements.

Kind regards,

Nicholas C. Manoukis

Academic Editor

PLOS ONE

Additional Editor Comments (optional):

Thank you for addressing final comments, and congratulations on completing a very nice study!
---

## [Editor Report · Acceptance letter]

17 May 2024

PONE-D-23-20467R2 

PLOS ONE

Dear Dr. Badolo, 

I'm pleased to inform you that your manuscript has been deemed suitable for publication in PLOS ONE. Congratulations! Your manuscript is now being handed over to our production team.

Kind regards, 

on behalf of

Dr. Nicholas C. Manoukis 

Academic Editor

PLOS ONE